# SHAPEWORLD: A new test methodology for multimodal language understanding

## Abstract

We introduce a novel framework for evaluating multimodal deep learning models with respect to their language understanding and generalization abilities. In this approach, artificial data is automatically generated according to the experimenter's specifications. The content of the data, both during training and evaluation, can be controlled in detail, which enables tasks to be created that require generalization abilities, in particular the combination of previously introduced concepts in novel ways. We demonstrate the potential of our methodology by evaluating a multimodal architecture on four different tasks, and show that our framework gives us insights into the model's capabilities and limitations.

## 1 Introduction

Deep learning methods have had a major impact on research in natural language processing and raised performance substantially in many of the standard evaluations. Moreover, multimodal tasks like image captioning (Karpathy and Li, 2015) or visual question answering (Antol et al., 2015) can now be tackled with great success. Such systems seem to solve the problems entirely on a sub-symbolic level, based only on raw image (and text) input, whereas previous approaches required a hand-crafted combination of various higher-level components.

There is, however, concern about how deep neural networks learn to solve such tasks. Investigations for image recognition (Szegedy et al., 2014; Nguyen et al., 2015; Zhang et al., 2017) have shown surprising behavior very different from what would be expected of their *"surpassing*

*human-level performance"* (He et al., 2015). Deep networks for language tasks may exhibit similarly odd behavior (Sproat and Jaitly, 2016; Arthur et al., 2016). Such results cast doubt on whether deep learning systems reliably acquire appropriate generalizations. However, given the recursive nature of language and the potentially enormous problem space of image captioning and similar tasks, acquiring the ability for reliable generalization will eventually be essential.

A more theoretical issue is the capability of network architectures to be able, in principle, to learn certain classes of structure. For instance, it has been shown that LSTMs possess the ability to handle long-range dependencies (Hochreiter and Schmidhuber, 1997; Gers and Schmidhuber, 2001). However, the formal experiments that have been done along such lines are limited, particularly in the multimodal domain of vision and language. While recent work indicates that the information encoded in image embeddings is rich enough for good captioning results, it is an open question whether current multimodal architectures are able, in principle, to combine visual information effectively to handle the full range of linguistic constructions.

This paper introduces a new test methodology for multimodal deep learning models. SHAPEWORLD is a framework for specifying datasets, which differ from standard evaluation datasets in two main ways. Firstly, a SHAPEWORLD dataset defines a process for generating artificial data, which is randomly sampled during training/testing according to constraints specified by the experimenter. Secondly, the evaluation focus of the methodology is on linguistic understanding capabilities of the type investigated by formal semantics. Hence the visual complexity and the open-class vocabulary size is reduced to a minimum, while allowing indefinitely complex syntactic con-

structions. Since it is possible to control data generation for both training and evaluation, we can introduce previously unseen configurations in the test data. This allows us to design tasks which require the system to recombine learned concepts to *understand* these novel instances, and is hence a form of zero-shot learning. We think of the SHAPEWORLD tasks as unit-testing multimodal systems for specific linguistic generalization capabilities, in a similar way to the bAbI task set of Weston et al. (2015) for text-only understanding.

We also present initial results on the performance of a generic multimodal architecture. The interest here is not obtaining high numbers, but rather analyzing the performance of a generic model with our evaluation methodology and demonstrating the potential of SHAPEWORLD in investigating multimodal deep neural networks. In fact, we have been surprised to see some poor performance even for simple tasks. We invite the community to explore the SHAPEWORLD tasks and use them to evaluate other successful image captioning or visual question answering systems. We want to emphasize, however, that the goal is not to achieve optimal performance by *tuning* a network architecture for one of these tasks. The central question is rather *whether* deep network architectures are able to successfully demonstrate the required understanding and generalization ability, and *whether* this will carry over to more complex tasks which can be defined using SHAPEWORLD.

## 2 Related work

With the increasing popularity of deep learning approaches, artificial data of various kinds is again seen as a valuable tool in experimentation. Recently, the simulation paradigm has been argued to be a promising driver for artificial intelligence research (Kiela et al., 2016). Various platforms following this paradigm have already been released, amongst others, OpenAI Gym (Brockman et al., 2016), DeepMind Lab (Beattie et al., 2016), Project Malmo (Johnson et al., 2016), to name a few of the most popular. An important advantage of simulated data is its infinite availability, particularly in light of the need of many deep learning models for huge amounts of data. Automatically generating data greatly reduces the cost, time and human effort. Moreover, it allows researchers to focus on particular problems, isolated from noisy and complex environments.

When focusing on language tasks, the simulation paradigm faces the problem that interesting language generation is a difficult task in its own right and that the difficulty increases with the complexity of the underlying world. The bAbI tasks (Weston et al., 2015) are generated by internally simulating a short scene and extracting a few simple sentences from it. A similar approach is taken by Narasimhan et al. (2015), but here the simulation is more complex, comprising a text-based role-playing game. The MazeBase game environment (Sukhbaatar et al., 2015) also uses language as a mean to represent the game world. However, the descriptions are in an abstract, formulaic format, and the focus of the simulation is much more on the planning than the language component. The long-term research proposal of Mikolov et al. (2015) also simulates a world where an agent learns to solve tasks by communication with a teacher module. At least for a start, this module is supposed to be scripted to automatically generate appropriate responses, given its internal knowledge of the world state.

Automatically generated data is common for tasks specifically focusing on the ability to efficiently process data of a certain formal structure. Here, data is deliberately stripped of any real-world connection to create an abstract capability check. Recent work in the context of deep learning has investigated sequence patterns (Joulin and Mikolov, 2015), combinatorial problems (Vinyals et al., 2015), or executing programming language code (Zaremba and Sutskever, 2014), amongst others. This kind of tasks is particularly common for neural network models (compare e.g., Bengio et al. (1994) more than twenty years ago). The reason for interest in abstract capability tests is that the learning process of deep neural networks is far more difficult to understand in a detailed way than shallower machine learning methods.

The multimodal tasks of image captioning and visual question answering are most closely related to our methodology, but usually consist of *repurposed* real-world photos and human-written descriptions[1]. However, there have been experiments in which parts of the data are artificial or generated automatically: for instance, automatic

---

[1]Although we use the term "real-world", in contrast to artificial simulations, it should be clear that it is very unlike the visual input which would be experienced by an entity situated in the real world

question generation (Ren et al., 2015) or modification of captions (Hodosh and Hockenmaier, 2016). Abstract Clipart scenes have been used for image captioning (Zitnick et al., 2016; Zitnick and Parikh, 2013). Most similar to our work are the experiments done by Bowman et al. (2015) and Sorodoc et al. (2016). Both generate artificial data fully automatically, based on abstract models, for a task which is targeted at a specific linguistic aspect, logical semantics and quantifiers, respectively.

Our own work is based on automatically generated, fully artificial data. This data, however, is not designed to address only a single structural problem, but is able to cover a whole range of linguistic phenomena. In fact, our generation system closely resembles classical work in formal semantics, where a statement corresponds to a logical expression which can be evaluated against an abstract world model (Montague, 1970). We utilize broad-coverage realization driven by the English Resource Grammar (ERG) (Flickinger, 2000) processed with Packard's Answer Constraint Engine (ACE, `http://sweaglesw.org/linguistics/ace/`). However, while SHAPEWORLD uses a logical representation internally, the *external* representation seen by the system under evaluation does not involve any abstract formalization of text or images. It nevertheless presents the intended problems clearly, without any uncontrolled noise, biases, or potentially hidden correlations, which can obfuscate results when using real-world images and text.

## 3 The SHAPEWORLD framework

The SHAPEWORLD framework[2] is based on the concept of **microworlds** – small and self-contained artificial scenarios – which guide the data creation process. The SHAPEWORLD microworlds simply consist of colored shapes. This closed-world domain allows for exhaustive coverage of the space of possible microworlds and associated captions, reducing the vocabulary size to (for now) far less than 100 words with an emphasis

---

[2] The SHAPEWORLD code is written in Python 3 and is available on GitHub (`https://git.io/vDWOX`). The generated data is returned as NumPy arrays, so that it is possible to integrate it into Python-based deep learning projects based on common frameworks like TensorFlow, Theano, etc. In our experiments, we use Tensorflow and we provide example scripts as part of the package. The Python package pydmrs (Copestake et al., 2016) is required for the internal DMRS-based caption generation.

on closed-class words. In the following we explain the details of the data generation process inside the SHAPEWORLD framework. A schematic illustration of the process is shown in figure 1.

### 3.1 Image caption agreement task

In this paper we focus on the task of **image caption agreement**. The system to be evaluated is presented with an image and a natural language caption, and has to decide whether they are consistent with each other. Compared to the classic image captioning task, it emphasizes understanding rather than the synthesis part of language use. We therefore avoid the problem of evaluating the appropriateness of a caption. The setup allows us to control the content of both modalities and consequently force a system to cope with difficult types of captions while obtaining a clear indicator of successful understanding.

One further motivation for the task is that human performance could be measured using the same setup. We would expect close-to-perfect human performance on the tasks described here, assuming time is not tightly constrained. Interesting comparisons are potentially possible where human performance depends on presentation: e.g., quantifiers such as *most* (Pietroski et al., 2009). However, we will not discuss this further in the current paper.

### 3.2 World and image generation

At the core of each abstract microworld instance lies an abstract **world model**. In the SHAPEWORLD framework, the internal representation of a microworld is simply a list of **entities** given as records containing their **primary attributes**, such as position, shape, color, which are considered to be high-level semantic aspects reflected in captions. In addition, an entity has secondary attributes and methods which control, for instance, details of visual appearance, visual noise infusion, or the collision-free placement of entities.

Currently, the SHAPEWORLD framework provides eight shape types (*square*, *rectangle*, *triangle*, *pentagon*, *cross*, *circle*, *semicircle* and *ellipse*) and seven colors (*red*, *green*, *blue*, *yellow*, *magenta*, *cyan* and *white*). The location and the secondary attributes are sampled uniformly or according to a truncated normal distribution (object size: 10% to 15% of image size; rotation: random; shade: shifted up to 50% towards black or white; and general noise: up to 10% deviation

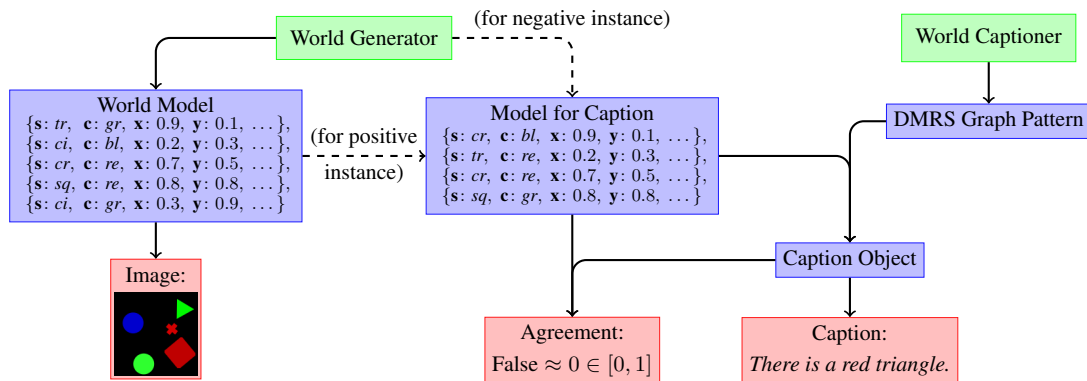

Figure 1: The generation process for image caption agreement data. Depending on whether a positive instance, i.e., a true statement about the world, or a negative instance is to be generated, either the first world model is used for caption generation or a new microworld is generated.

per pixel). Importantly, all these ways of infusing noise can be controlled, which is useful particularly since noise is often seen as important for successful training of deep models.

The **generator module** automatically generates a world model by randomly sampling all these attributes from a set of available values. Both the values and other aspects of the generation process can be specified and adjusted appropriately. The internal abstract representation is then used as a basis to extract a concrete microworld instance consisting of image and caption. An image (of size $64 \times 64$ in this work) is just a straightforward visualization of the world model. Note that the exact visual appearance of an entity with certain primary attributes varies from instance to instance. Caption generation is more complex, as discussed in the next section.

### 3.3 Caption generation

The caption interface specifies the methods a language generation module has to provide to integrate into the data generation process. We currently provide an implementation using a grammar-based approach. More specifically, Dependency Minimal Recursion Semantics (DMRS) is an abstract semantic graph representation designed for use with high-precision grammars, such as those distributed by the DELPH-IN consortium. A semantic representation like DMRS is particularly suited for the SHAPEWORLD framework, since it essentially mirrors parts of the internal world model and hence acts like a language-specific annotation.[3] For instance, noun nodes

correspond to entities, adjective nodes add attributes, and verb phrase nodes or sub-graphs specify relations between entities. Below an example of a DMRS semantic graph with its compositional components colored:

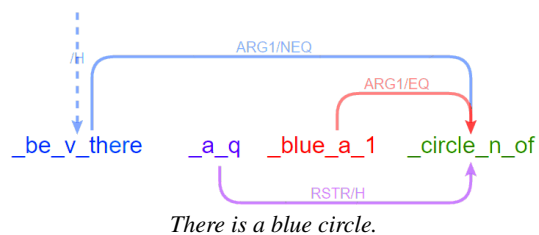

*There is a blue circle.*

Compositionality of the semantic representation is a useful property and an important reason for our choice of using DMRS. Given compositionality, it is enough to specify the semantics of words – or, more precisely, of the linguistic atoms in the SHAPEWORLD context, which potentially are sub-graphs with multiple nodes and inner link structure – to be able to obtain the corresponding semantics of composed sub-graphs and so generate a wide range of combinatorially different captions. For instance, the semantics of words like *square* or *red* iteratively filter a subset of agreeing entities, transitive relations like *to the left of* act similarly on pairs of entity sets, and quantifiers compare the cardinality of two entity sets.

Figure 2 shows an example of a more complex compositional caption, which contains the DMRS graph above as sub-graph. It also illustrates how various details are automatically inferred by the ERG, including number-agreement between subject and verb, and between quantifier and noun, and realization of an adjective as relative clause.

---

[3] Although we currently use the English Resource Grammar (Flickinger, 2000), other DELPH-IN grammars use a compatible approach, so SHAPEWORLD could be ported to other languages relatively easily.

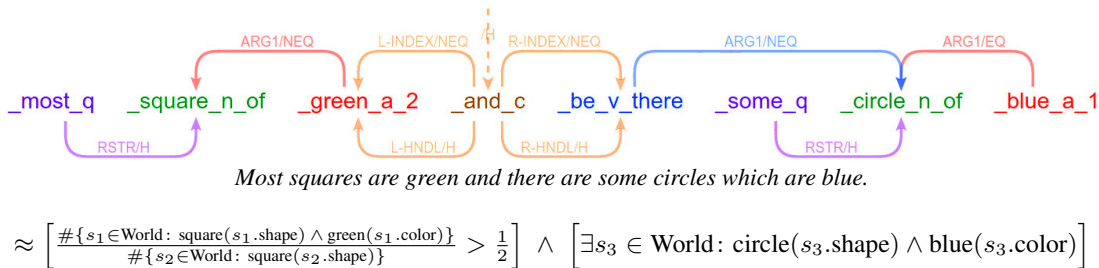

*Most squares are green and there are some circles which are blue.*

$$\approx \left[ \frac{\#\{s_1 \in \text{World}: \text{square}(s_1.\text{shape}) \land \text{green}(s_1.\text{color})\}}{\#\{s_2 \in \text{World}: \text{square}(s_2.\text{shape})\}} > \frac{1}{2} \right] \land \left[ \exists s_3 \in \text{World}: \text{circle}(s_3.\text{shape}) \land \text{blue}(s_3.\text{color}) \right]$$

Figure 2: An example of a DMRS graph corresponding to a more complex caption, with compositional components colored. The logical formula gives the formal semantic interpretation over a world model.

This greatly facilitates the generation of a combinatorially large amount of captions and makes the DMRS graph patterns reusable, as outlined in section 4. Figure 2 furthermore gives a formal semantic interpretation of the caption meaning as logical formula over a world model. In essentially the same way, the agreement of a caption with a microworld is computed in the SHAPEWORLD framework.

Similar to the generator module, the **captioner module** also randomly samples from a set of dataset-specific **DMRS graph patterns**, which are then applied to a world model to construct an *agreeing* **caption object**. Such a caption object can be turned into natural language. Its ability to check whether another world model would agree with the semantics of this caption is important for the generation of negative instances, i.e., captions that do not agree with the corresponding microworld they are supposed to describe. These captions are obtained by sampling a second, *false* world model, extracting a caption object from it, and ensuring that it does not accidentally also agree with the first, *true* microworld.

### 3.4 Training and testing on SHAPEWORLD datasets

Since SHAPEWORLD datasets are actually data generation processes, training and evaluation work differently from classic datasets. Where usually one has a fixed set of test instances, here models are trained and tested on a fixed set of more abstract **configuration constraints**. In particular, in our experiments, the constraints for testing (and validation) differ from the training constraints, hence requiring true generalization abilities. This means that, for instance, a certain shape-color-combination, a specific number of objects, a spatial location or a caption type can be held-out and never generated during training. The model is presented with instances of this configuration only when evaluated and is hence required to recombine concepts it acquired from the training data. It is thus possible for a system to achieve optimal performance during training, but completely fail the evaluation. Another parameter is the ratio of correct instances, which may differ between training and evaluation. This adds yet another level of control to allow us to analyze the behavior of a model.

## 4 Some SHAPEWORLD datasets

In this paper we look at four datasets, each designed to investigate an aspect of the capability to understand language in a multimodal setup and to generalize to new instances not seen during training. Two example images of microworlds for each dataset are shown in Figure 3.

**OneShape** The first and simplest dataset requires the evaluated system to learn to separate the concepts of shape and color, instead of treating each combination as an atomic object. For this, microworlds are generated which contain a single object, together with a simple existential statement of the form: *There is a [color] [shape]*. One color-shape-combination, *red square*, is reserved for evaluation, which leaves 55 combinations for training.

**MultiShape** This dataset generates worlds with up to four objects, with the same type of existential statement as before, describing one of these objects. The system is evaluated on worlds containing five objects, requiring the ability to focus attention on one object without being distracted by other objects.

**Spatial** Microworlds with two objects are generated in this dataset accompanied by a statement about their relative location: *to the left of*, *to the*

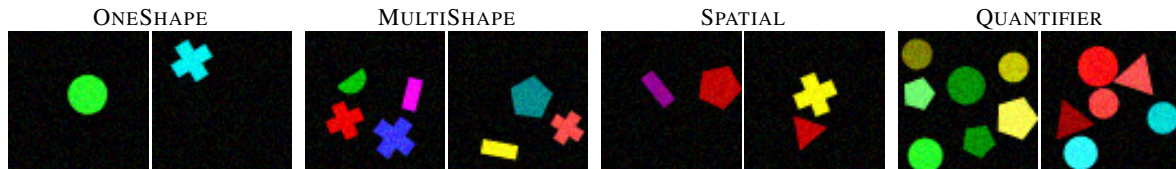

Figure 3: Two example microworlds generated by each of the four presented SHAPEWORLD datasets.

*right of*, *above* or *below*. The overall caption is of the form: *The [color] [shape] is [spatial] the [color] [shape]*. As in the first dataset, a previously unseen attribute combination is presented for evaluation.

**Quantifier** The quantifier dataset generates microworlds of three to six objects, with seven used for evaluation. The captions focus on the quantifiers *no*, *a*, *the*, *some*, *two*, *most*, *all* and *every*. Compatibility with the caption is analyzed according to the quantifier's usual cardinality-based interpretation in formal semantics. For instance, *most* is considered to be true if the **target** attribute applies to more than half of the **domain** set of objects. There are a variety of possible caption patterns in this dataset with different degrees of underspecification in domain and target, as illustrated in the examples below:

- *Some shapes are green.*
- *Most shapes are rectangles.*
- *No shape is a red triangle.*
- *All triangles are green.*
- *Two blue shapes are pentagons.*

Since we first sample a microworld model and subsequently a caption, here we cannot easily control the sampling process to uniformly sample each possible caption, as is trivially the case for the other datasets. Consequently, we restrict the number of different shapes and colors possible in a world, so that situations for the quantifiers *all*, *every* and *most* become more likely. Moreover, we adapt the probabilities of the sampling process so that we observe a more uniform distribution over caption patterns when sampling a large number of instances.

An important property of the SHAPEWORLD datasets is their *compositionality*. Instead of having to define a dataset from scratch every time, we can specify **atomic datasets** like the ones described above, and then combine them in a **mixer dataset**, which tests for various different aspects

of multimodal language understanding simultaneously. Reusability in fact applies even further down in the component hierarchy. For instance, we use the same generic world generator module for all four datasets. This is even more important for caption generation where, for instance, a logical combinator dataset can reuse different world captioner modules to generate simple statements which then are merged by logical connectives.

## 5 Experiments

As discussed in the introduction, the aim of our experiments is to demonstrate that SHAPEWORLD allows detailed investigation of neural network architectures, rather than high performance as such, although naturally a certain minimum performance is required for the results to be interesting.

### 5.1 Network architecture

We evaluate a generic multimodal deep neural network architecture shown in figure 4, which is inspired by recent work on visual question answering (Antol et al., 2015). The visual input is processed by a convolutional neural network (CNN) module (LeCun et al., 1989) with three layers of convolutions (filter sizes: 5, 3, 3; number of filters: 16, 32, 64) followed by a 2×2 max-pooling step, or average-pooling in the last layer, respectively. With an image size of 64×64, the obtained world embedding has 4096 dimensions. The textual input is transformed to 32-dimensional embeddings and passed on to a long short-term memory (LSTM) module (Hochreiter and Schmidhuber, 1997) with state size 64. The final state is used as the caption embedding, which is scaled and fused with the world embedding via pointwise multiplication. After a fully-connected hidden layer of 512 dimensions, yield the agreement score. The entire architecture is trained end-to-end on the task. Both the CNN module and the word embeddings are included in the training, as opposed to using pre-trained, general-purpose versions.

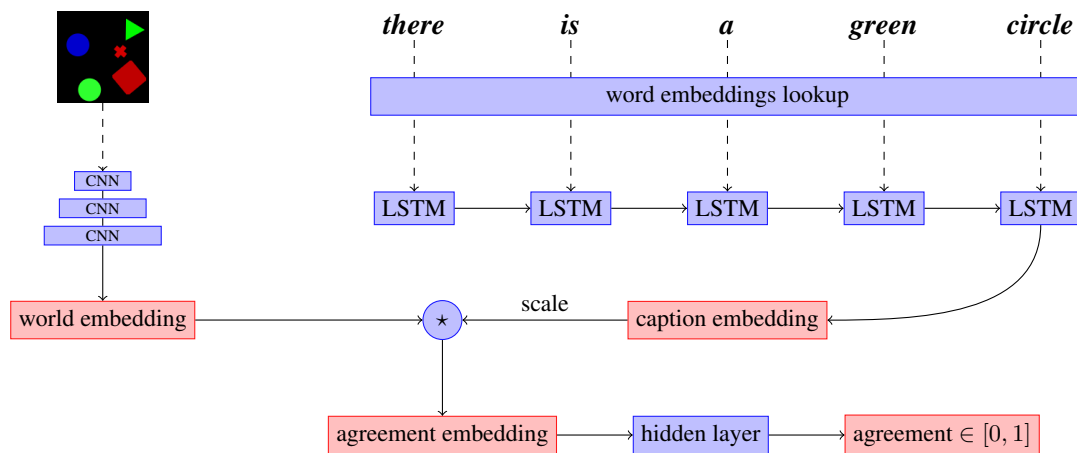

Figure 4: A schematic depiction of the multimodal deep neural network which we evaluate on the SHAPEWORLD datasets.

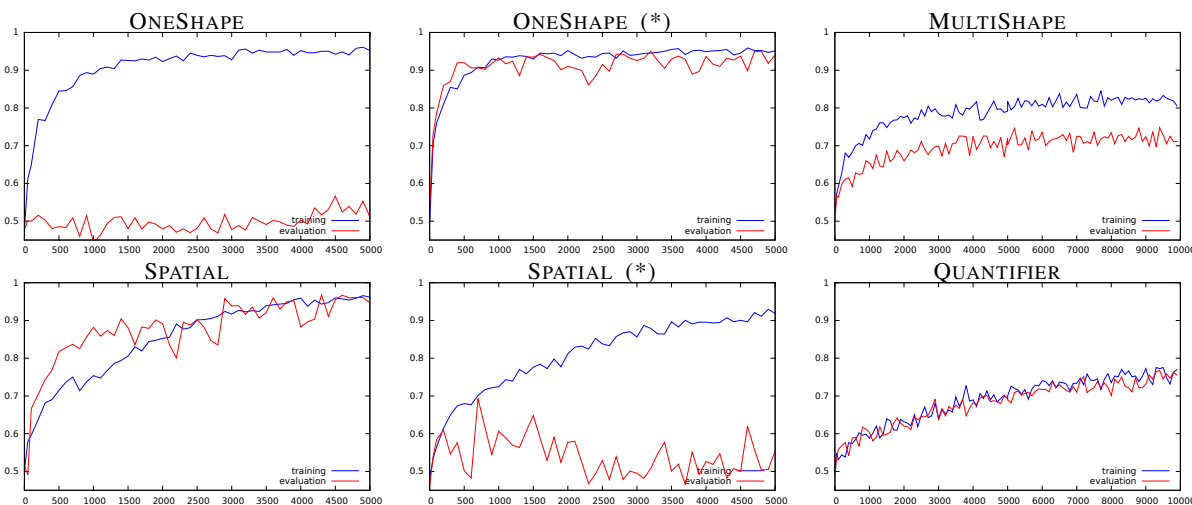

Figure 5: Performance of the evaluated multimodal deep neural network on the four SHAPEWORLD datasets, measured as accuracy of correctly judged image caption agreements. .

## 5.2 Training and evaluation setup

In all experiments, we use a training batch size of 128 where the ratio of correct instances is 33%. To track the performance of the model throughout the learning process (as opposed to just at the end), we add an evaluation step every 100th training iteration, where we obtain the accuracy on *both* training *and* evaluation instances. These numbers are calculated on basis of a batch of 1024 instances each, with a correct instance ratio of 50%, and are used to visualize and investigate the learning performance in our experiments.

## 5.3 Results

Figure 4 shows the performance results of our experiments. We observe a high accuracy on the ONESHAPE dataset on training instances, but an unexpectedly low and unstable performance on the

held-back instances. This indicates that the system does not naturally learn to distinguish shape and color as two separate concepts, but rather learns to *classify* each shape-colour-combination individually – after all, the ONESHAPE dataset essentially resembles a classification task. In a followup experiment, we reduced the embedding size to 8 and the LSTM state size to 16 dimensions (marked with (*) in figure 4). Being given less *memory* to learn simple non-generalizing classification, the model indeed achieves similarly high accuracy numbers in evaluation as during training.

The training performance on the MULTISHAPE dataset is relatively low and plateaus at around 80-85% after roughly 5000 iterations. The evaluation accuracy, in comparison, is not that much lower, indicating that the generalization to an unseen number of objects is learned, to some degree.

On the SPATIAL dataset, unlike on ONESHAPE, the system successfully learns the shape-color-distinction from the start. Note that each caption here involves two object descriptions, so instead of simply learning classification, it is necessary to disentangle the words of the caption. The fact that the evaluation accuracy is higher than the training accuracy in the beginning is because the held-back combination *red square* is (unsurprisingly) easier to recognize than some others. Again, we also ran an experiment with reduced embedding and LSTM size. Contrary to ONESHAPE, the accuracy curves indicate that this impedes the learning of an appropriate generalization ability for the more complex SPATIAL task.

The QUANTIFIER dataset confirms the lower accuracy of captions focusing on *parts* of a world which contains multiple shapes. The training accuracy is around 5% lower than for MULTISHAPE and shows no substantial difference to the validation accuracy, implying that quantifiers, expectedly, are more difficult to learn than existential statements, but definitely possible to some degree.

## 5.4 Discussion

We want to emphasize the fact that the training accuracy here represents an interesting measure on its own. Contrary to the common evaluation methodology, where training performance is less expressive after multiple iterations over the training data, each instance is randomly generated anew in our setting. Consequently, even the training accuracy reflects a learning process, and *overfitting* happens on a conceptual/structural level instead of a numerical correlation level. Investigations of overfitting in this sense are not easily possible with non-artificial data.

Another type of investigation of what a model has learned involves manipulation of the evaluation data. For instance, we analyzed the fact that the network only achieves around 95% accuracy on the supposedly simple ONESHAPE dataset. Looking at a few misclassifications, we found that sometimes the negative instances are problematic where either only the shape or only the color attribute differed from what the caption stated. We thus created a modified version of the ONESHAPE dataset which generates only negative instances of this structure. Applying the trained model to this dataset indeed gave a training accuracy of around 80% and an evaluation accuracy of only 55-60%.

Changing the network parameters altered the performance for the ONESHAPE and SPATIAL datasets in interestingly different ways. In general, when we changed parameters in our experiments, we often observed either no learning at all, less (or no) improvement in evaluation accuracy compared to training accuracy, or similar curves for both numbers. Given our aim to set up capability unit-tests, such discrete, to some degree invariant outcomes are what we ideally expect – a behavior we intend to investigate further. Another aspect, which might turn out interesting in more complex experiments is the shape of the accuracy curves.

## 6 Conclusion and future work

We have presented a new test methodology and framework, SHAPEWORLD, for multimodal deep learning models with a focus on formal-semantic style generalization capabilities. In this framework, artificial data is automatically generated, and datasets specify parameters of this generation process to create appropriate data, which targets a specific multimodal understanding task. We evaluated a neural network architecture on four image caption agreement datasets, where the system has to decide whether a statement applies to an image. We show how the SHAPEWORLD framework can be used to investigate the learning process of deep neural networks, and that it clearly indicates whether certain abilities are acquired. There are clearly some deficiencies in the generalization ability of the evaluated network – one aim of this paper is to invite the research community to evaluate their architectures on our datasets.

The SHAPEWORLD framework is still under development. In particular, we plan to add new datasets addressing other aspects of language, as well as integrating options to enhance the language generation module, with the aim of providing more varied and natural image descriptions. One option would be to integrate a subsequent step applying paraphrase rules after caption generation – Copestake et al. (2016) describe how this can be implemented on the level of DMRS graphs. In addition, the framework can easily be extended to produce more complex worlds, e.g., based on Cliparts (Zitnick et al., 2016). However, we plan to stick to the simplicity and abstractness of our microworlds for now, since we think they offer several advantages and enough richness for various interesting investigations.

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
