# Peer review of "ShapeWorld: A new test methodology for multimodal language understanding"

_ACL 2017 — decision unknown_

[Official Review · Reviewer 1 · rating 2 · confidence 4]
soundness 3 · originality 3 · clarity 4 · impact 3 · substance 2 · appropriateness 5 · meaningful comparison 3 · presentation format Oral Presentation

This paper proposes a method for generating datasets of pictures from simple
building blocks, as well as corresponding logical forms and language
descriptions.
The goal seems to be to have a method where the complexity of pictures and
corresponding desciptions can be controlled and parametrized. 

 - The biggest downside seems to be that the maximally achievable complexity is
very limited, and way below the complexity typically faced with
image-captioning and other multimodal tasks. 
 - The relative simplicity is also a big difference to the referenced bAbI
tasks (which cover the whole qualitative spectrum of easy-to-hard reasoning
tasks), whereas in the proposed method a (qualitatively) easy image reconition
task can only be quantitatively made harder, by increasing the number of
objects, noise etc in unnatural ways.
 - This is also reflected in the experimental section. Whenever the
experimental performance results are not satisfying, these cases seem like
basic over/underfitting issues that may easily be tackled by
restricting/extending the capacity of the networks or using more data. It is
hard for me to spot any other qualitative insight.
 - In the introduction it is stated that the "goal is not too achieve optimal
performance" but to find out whether "architectures are able to successfully
demonstrate the desired understanding" - there is a fundamental contradiction
here, in that the proposed task on the one side is meant to provide a measure
as to whether architectures demontrate "understanding", on the other hand the
score is not supposed to be taken as meaningful/seriously.

General comments:
The general approach should be made more tangible earlier (i.e. in the
introction rather than in section 3)

[Official Review · Reviewer 2 · rating 2 · confidence 5]
soundness 3 · originality 3 · clarity 4 · impact 3 · substance 3 · appropriateness 5 · meaningful comparison 3 · presentation format Poster

- Strengths:

The authors introduce a new software package called ShapeWorld for
automatically generating data for image captioning problems. The microworld
used to generate the image captions is simple enough to make the data being
generated and errors by a model readily interpretable. However, the authors
demonstrate that configurations of the packages produce data that is
challenging enough to serve as a good benchmark for ongoing research.

- Weaknesses:

The primary weakness of this paper is that it does look a bit like a demo
paper. The authors do provides experiments that evaluate a reasonable baseline
image captioning system on the data generated by ShapeWorld. However, similar
experiments are included in demo papers.

The paper includes a hyperlink to the software package on github that
presumably unmasks the authors of the paper.

- General Discussion:

Scientific progress often involves some something analogous to vygotsky's zone
of proximal development, whereby progress can be made more quickly if research
focuses on problems with just the right level of difficulty (e.g., the use of
tidigits for speech recognition research in the early 90s). This paper is
exciting since it offers a simple microworld that is easy for researchers to
completely comprehend but that also is just difficult enough for existing
models.

The strengths of the work are multiplied by the fact that the software is
opensource, is readily available on github and generates the data in a format
that can be easily used with models built using modern deep learning libraries
(e.g., TensorFlow).

The methods used by the software package to generate the artificial data are
clearly explained. It is also great that the authors did experiments with
different configurations of their software and a baseline image caption model
in order to demonstrate the strengths and weakness of existing techniques. 

My only real concern with this paper is whether the community would be better
served by placing it in the demo section. Publishing it in the non-demo long
paper track might cause confusion as well as be unfair to authors who correctly
submitted similar papers to the ACL demo track.